# Impact of Acid Hydrolysis on Morphology, Rheology, Mechanical Properties, and Processing of Thermoplastic Starch

**DOI:** 10.3390/polym17101310

**Published:** 2025-05-11

**Authors:** Saffana Kouka, Veronika Gajdosova, Beata Strachota, Ivana Sloufova, Radomir Kuzel, Zdenek Stary, Miroslav Slouf

**Affiliations:** 1Institute of Macromolecular Chemistry of the Czech Academy of Sciences, Heyrovsky Sq. 2, 16206 Prague, Czech Republic; kouka@imc.cas.cz (S.K.); gajdosova@imc.cas.cz (V.G.); beata@imc.cas.cz (B.S.); stary@imc.cas.cz (Z.S.); 2Department of Physical and Macromolecular Chemistry, Faculty of Science, Charles University, Hlavova 2030, 12840 Prague, Czech Republic; ivana.sloufova@natur.cuni.cz; 3Faculty of Mathematics and Physics, Charles University, Ke Karlovu 5, 12116 Prague, Czech Republic; radomir.kuzel@matfyz.cuni.cz

**Keywords:** thermoplastic starch, low viscosity, melt mixing, processing temperature

## Abstract

We modified native wheat starch using 15, 30, and 60 min of acid hydrolysis (AH). The non-modified and AH-modified starches were converted to highly homogeneous thermoplastic starches (TPSs) using our two-step preparation protocol consisting of solution casting and melt mixing. Our main objective was to verify if AH can decrease the processing temperature of TPS. All samples were characterized in detail by microscopic, spectroscopic, diffraction, thermomechanical, rheological, and micromechanical methods, including in situ measurements of torque and temperature during the final melt mixing step. The experimental results showed that (i) AH decreased the average molecular weight preferentially in the amorphous regions, (ii) the lower-viscosity matrix in the AH-treated starches resulted in slightly higher crystallinity, and (iii) all AH-modified TPSs with a less viscous amorphous phase and higher content of crystalline phase exhibited similar properties. The effect of the higher crystallinity predominated at a laboratory temperature and low deformations, resulting in slightly stiffer material. The effect of the lower viscosity dominated during the melt mixing, where the shorter molecules acted as a lubricant and decreased the in situ measured processing temperature. The AH-induced decrease in the processing temperature could be beneficial for energy savings and/or possible temperature-sensitive admixtures for TPS systems.

## 1. Introduction

Starch is the primary carbohydrate reserve in plants, composed of two polysaccharides: amylose and amylopectin [1]. Amylose is a linear polymer linked by α-1,4 glycosidic bonds, while amylopectin is formed by highly branched molecules with α-1,4 linkages in the backbone and α-1,6 linkages at the branching points [2]. Molecular weights of both polymers are very high: ca. 10^6^ g/mol for amylose and ca. 10^8^ g/mol for amylopectine [3,4,5,6]. In the native starch granules, amylose is localized in the amorphous regions, whereas amylopectin is in semicrystalline regions [7,8]. Due to its high molecular weight, the native starch cannot be processed by melt mixing like other common polymers, because it decomposes before melting [9]. To overcome this limitation, starch is typically mixed with low-molecular-weight plasticizers such as water, glycerol, citric acid, and/or urea [10,11]. In our previous work, we demonstrated that oligomers such as maltodextrin can be added as lubricants that decrease the starch processing temperature [2].

When starch is heated in the presence of plasticizers and subjected to shear forces, it undergoes gelatinization—a process in which the starch granules are partially disrupted and merged [8,12]. This transforms the native starch to thermoplastic starch (TPS), which can be processed similarly to conventional thermoplastic polymers. TPS can be prepared by solution casting in the excess of water (SC; [13]), by melt mixing (MM; [14]), or by the combination of both protocols (SC+MM; [15]). TPS must be prepared at elevated temperatures, as starch gelatinization generally occurs above 100 °C, regardless of the water content [16]. However, even a moderate reduction in the processing temperature would be advantageous for energy-saving reasons. One of possible strategies to decrease the processing temperature is the lowering of starch molecular weight, *M*. Polymers with lower *M* exhibit lower viscosity and require lower temperatures to flow [17].

The reduction in the starch molecular weight can be achieved through physical methods, such as hydrothermal treatment [18] or ultrasonication [19,20], by means of chemical processes (typically acid hydrolysis [21]), or with enzymatic methods [22]. Acid hydrolysis (AH) is widely employed in the food industry to tailor starch’s physicochemical properties [23,24]. It is often conducted below the gelatinization temperature to preserve the granular structure of starch [25]. The AH proceeds in two phases: an initial rapid hydrolysis targeting the amorphous regions, followed by a slower phase where crystalline regions are hydrolyzed [26]. Consequently, acid-hydrolyzed starch often exhibits higher crystallinity than native starch [23,25]. The acid hydrolysis of starch also affects the rheological properties of the TPS: mild acid hydrolysis can enhance gel strength and stiffness, while prolonged hydrolysis reduces molecular weight, producing weaker gels [27]. Although the acid hydrolysis of starch is a well-established industrial process, particularly in the context of food applications [23,28], limited information is available on its role in plasticization and the resulting processing properties of thermoplastic starch prepared by melt mixing [29,30].

In this contribution, we subjected native wheat starch to acid hydrolysis (AH) for 0, 15, 30, and 60 min. The AH starch was thermoplasticized using single-step solution casting (SC) and a two-step protocol combining solution casting with melt mixing (SC+MM), which yields highly homogeneous thermoplastic starch [15]. The prepared materials were characterized thoroughly by numerous microscopic, spectroscopic, diffraction, rheological, thermomechanical, and micromechanical measurements. The first question we asked was how the AH treatment influences the morphology, homogeneity, and mechanical performance of thermoplastic starch. The second question was if the acid hydrolysis can decrease the starch viscosity and, as a result, its processing temperature during melt mixing. Highly homogeneous starch with a lower processing temperature would be a promising material for both technical applications in packaging and agriculture (energy savings during the processing) and medical applications in pharmaceutics (the high homogeneity to secure reproducibility and lower processing temperature to improve the stability of sensitive admixtures, such as antibiotics [31]).

## 2. Materials and Methods

### 2.1. Materials

The wheat starch powder used in this study (starch type A, amylose content ca. 25%) was supplied by Škrobárny Pelhřimov a.s. (Pelhřimov, Czech Republic). Anhydrous glycerol (C_3_H_8_O_3_; >99%), hydrochloric acid, (HCl; 35%), and sodium bromide (NaBr; >99%) reagents were bought from Lach-Ner s.r.o. (Neratovice, Czech Republic).

### 2.2. Preparation of Hydrolyzed Starch

Acid hydrolysis was performed following a previously described method [32] with a few minor modifications. A 40% starch slurry was prepared by dispersing starch in an aqueous 1M hydrochloric acid solution. The reaction was conducted under mechanical stirring and mild heating in a water bath maintained at 45 °C. Hydrolysis durations were varied (15 min, 30 min, and 60 min) to evaluate the effects of reaction time. To terminate the reaction, the mixture’s pH was adjusted to 7 using a 1 M sodium hydroxide solution. The neutralized slurry was then washed thoroughly with distilled water, followed by filtration. The resulting material was dried in an oven at 45 °C for 24 h and subsequently cooled to room temperature and milled into a fine powder.

### 2.3. Preparation of Thermoplastic Starch

The thermoplastic starch (TPS) was prepared from the native starch powders (S) with various acid hydrolysis (AH) times. TPS was prepared by both a single-step solution casting protocol (SC; Section 2.3.1) and two-step protocol comprising solution casting followed by melt mixing (SC+MM; Section 2.3.2). This approach is based on our previous work [15], with slight modifications described below. The TPS samples prepared in this study are summarized in Table 1.

#### 2.3.1. TPS Prepared by Single-Step Solution Casting

The starch powders (70 wt.%) were premixed with glycerol (30 wt.%) and distilled water (6 parts of water per 1 part of starch) with a magnetic stirrer in a beaker for 30 min at room temperature. The pre-mixed suspension was transferred to a mechanical stirrer, where it was heated to initiate the starch gelatinization. A significant increase in viscosity was observed at temperatures between 63 °C and 70 °C. The viscosity increase indicated the onset of gelatinization, and the mixture was stirred continuously for about 15 min until a homogeneous pudding-like consistency was obtained. Then, the solution was cast onto a polyethylene (PE) foil to form a film with a thickness of approximately 2 mm. The thin film was left to dry at room temperature for three days to allow for the evaporation of residual water.

#### 2.3.2. TPS Prepared by Two-Step Protocol: Solution Casting Followed by Melt Mixing

The solution-casted and dried TPS films from the previous steps were processed by melt mixing using a twin-screw laboratory kneader (Brabender Plasti-Corder, Duisburg, Germany) to further increase their homogeneity [15,33]. The samples were mixed in the chamber pre-heated to 120 °C, using a rotation speed of 60 rpm for at least 8 min while recording the real processing temperature and torque moments. Subsequently, the sample was compression molded into plaques with a thickness of 2 mm. This was achieved using a laboratory hot press (Fontijne Grotnes; Vlaardingen, The Netherlands) in the following multi-step process: Initially, the material was pressed at 130 °C under a pressure of 50 kN for 2 min to deaerate. This was followed by pressing at the same temperature under 100 kN for 2 min. Finally, the press with the molded plaques was cooled with water while maintaining a pressure of 100 kN for approximately 10 min, until room temperature was reached.

#### 2.3.3. Storing of the Samples in Defined Conditions

The final TPS plaques were stored in defined conditions: at room temperature in a desiccator over a supersaturated solution of sodium bromide, which yielded a relative humidity = 57%. The samples were in the desiccator all the time, being removed just before the measurements of their properties by the characterization methods described below. Our experience showed that storing the samples at a well-defined humidity leads to more reproducible results of mechanical and rheological measurements, even if storage at ambient conditions is possible as well.

### 2.4. Characterization Methods

#### 2.4.1. Light and Electron Microscopy

The morphology and homogeneity of TPS samples were checked by light microscopy (LM), polarized light microscopy (PLM), and scanning electron microscopy (SEM). The thin sections (thickness 5 μm) for LM and PLM were prepared with a rotary microtome RM 2255 (RM 2255; Leica, Vienna, Austria). The sections were placed in a thin oil layer between the microscopic glasses and observed in transmitted light (LM) or polarized transmitted light (PLM) under a Nikon Eclipse 80i microscope (supplied by Laboratory Imaging, Prague, Czech Republic). The fracture surfaces for SEM observations were prepared in liquid nitrogen. The specimens were fixed on a metallic support with a conductive adhesive carbon tape (Plano GmbH, Wetzlar, Germany), sputter-coated with a thin platinum layer (vacuum sputter coater SCD 050; Leica, Austria; thickness of the Pt layer: approx. 4 nm), and observed under a MAIA3 SEM microscope (Tescan, Brno, Czech Republic) using secondary electron imaging at an accelerating voltage of 3 kV.

#### 2.4.2. Vibrational Spectroscopy

Fourier-transform infrared (FTIR) spectra were recorded on a Thermo Fisher Scientific Nicolet iS50 FTIR spectrometer (Nicolet CZ s.r.o.; Prague, Czech Republic) using a 4 cm^−1^ resolution in the 400–4000 cm^−1^ region (with KBr beamsplitter and Happ-Genzel apodization) by means of the ATR (diamond crystal) technique. Standard ATR correction was applied. Raman spectra were collected on a dispersive micro Raman system, MonoVista CRS+ (Spectroscopy & Imaging GmbH; Warstein, Germany), interfaced to an Olympus microscope (50× objective) equipped with a 785 nm excitation laser, grating of 150 g/mm, spectrograph aperture with a 50 µm slit, and laser power of 10.5 mW on the sample. The wavelength and intensity calibration of the spectrometer was performed by the software-controlled auto alignment procedure using mercury and Ne-Ar lamps. The total amount of 150–300 spectra for each sample with an exposure time of 5 s per spectrum was collected. All spectra were subsequently baseline corrected according to the singular value decomposition method [34], averaged, and normalized (max–min).

#### 2.4.3. Wide-Angle X-Ray Scattering

Wide-angle X-ray scattering (WAXS) patterns were obtained with a Panalytical MPD system (Panalytical; Almelo, the Netherlands) with a vertical goniometer, CoKα radiation, variable divergence slits (mostly fixed irradiated length of 10 mm), and 1D Pixcel detector. The specimens (powders, films, bulk samples) were always placed on so-called non-diffracting Si substrates, giving little background. The ranges of 4–70 deg 2θ (CoKα) were taken with the total measurement time of 1 hr. For comparison with the other literature, the final diffractograms were recalculated so that they corresponded to more common CuKα radiation (the wavelengths of CoKα and CuKα were taken as 1.79 and 1.54 Å, respectively). The crystallinity (weight fraction of crystalline phase) was calculated by means of Fityk software [35].

#### 2.4.4. Dynamic Mechanical Thermal Analysis

The thermomechanical properties of the TPS systems were measured in torsion by dynamic mechanical thermal analysis (DMTA) on specimens of a rectangular platelet shape (40 mm × 10 mm × 2 mm), using an ARES G2 (TA Instruments, New Castle, DE, USA) in oscillatory mode, at a deformation frequency of 1 Hz. The deformation amplitude ranged from 0.01 to 3% (regulated automatically by the auto-strain function, in response to sample resistance). The investigated temperature range was from −90 to 140 °C, while the heating rate was 3 °C/min. The temperature dependences of the storage shear modulus (G′), =the loss modulus (G″), and the loss factor tan (δ) were recorded.

#### 2.4.5. Rheology

Rheological properties of the TPS systems were measured in shear on a strain-controlled ARES G2 rheometer (TA Instruments, New Castle, DE, USA) using a parallel plate fixture with a diameter of 30 mm (plates with cross-hatched surface to prevent slipping). The thickness of the specimens was 2 mm. At first, the linear viscoelasticity region (LVER) was determined in view of the dependence of the storage modulus on the strain amplitude, which was measured at 120 °C at a frequency of 1 Hz. Next, the frequency sweep experiments were performed in a frequency range from 0.1 to 100 rad/s at a strain amplitude of 0.05% (always well within the LVER) and at a constant temperature of 120 °C. To ensure a uniform temperature in the specimen, all samples were equilibrated for 2 min prior to the start of each type of experiment. The frequency sweep was performed twice for each TPS specimen.

#### 2.4.6. Microindentation Hardness Testing

Micromechanical properties were measured with an instrumented microindentation hardness tester (MCT tester; CSM, Switzerland). The microindentation hardness testing (MHI) experiments were carried out using a Vickers method: a diamond square pyramid (with an angle between non-adjacent faces of 136°) was forced against the flat surface of a specimen. The flat smooth surfaces for the testing were prepared by cutting from the 2 mm thick plates with a rotary microtome RM 2255 (Leica, Vienna, Austria). The micromechanical properties were deduced from the loading force, which was measured as a function of penetration depth. From each sample, three independent cut surfaces were prepared. For each measured surface, at least 10 independent measurements/indentations were performed and the final results were averaged. As we repeated the measurement for each sample three times to verify the reproducibility, the final averaged values of all micromechanical properties represent more than 90 measurements (3 cut surfaces per sample × at least 10 indentation per surface × 3 repetitions = more than 90). The parameters of MHI measurements were as follows: maximal loading force *F*_max_ = 500 mN, dwell time (time of maximal load) 60 s, and linear loading and unloading rates 15,000 mN/min (i.e., ~2 s to achieve and release *F*_max_). The evaluated micromechanical properties were as follows: an indentation modulus (*E*_IT_) proportional to the macroscopic elastic modulus, an indentation hardness (*H*_IT_) proportional to the macroscopic yield stress, Martens hardness (*H*_M_) also referred as universal hardness, indentation creep (*C*_IT_) related to the macroscopic creep, and the elastic part of the indentation work (*η*_IT_) defined as the ratio of elastic deformation to total deformation. The calculations of *E*_IT_ and *H*_IT_, *E*_IT_ were based on the theory of Oliver and Pharr [36], while the values of *H*_M_, *C*_IT_, and *η*_IT_ were independent of the O&P theory [37]. The exact definitions of the above-listed micromechanical properties can be found in textbooks on micromechanical properties [38,39], and more detailed descriptions of the MHI measurements are also given in our recent studies [40,41,42].

#### 2.4.7. In Situ Measurements During Melt Mixing

During the melt mixing of TPS in a laboratory kneader (Section 2.3.2), we recorded the values of torque (TQ, moment of force; in Nm) and real processing temperature (*T*; in °C) as a function of processing time (*t*; in seconds). To ensure the maximal reproducibility of the measurements, the kneading chamber was filled with the same amount of material (75 g) at the same time (2 min). Moreover, the final values of TQ and *T* were evaluated from the final part of the TQ-*t* and *T*-*t* curves, after ca. 6 min, when the curves reached a plateau indicating that the mixed system achieved a steady state.

## 3. Results and Discussion

### 3.1. Morphology and Homogeneity

#### 3.1.1. Light and Electron Microscopy

Figure 1 displays representative polarized-light micrographs (PLMs) of thermoplastic starches prepared by two different protocols (one-step SC vs. two-step SC+MM) with or without acid hydrolysis (AH). In the PLMs, the bright areas indicate anisotropic material. In the case of TPS materials, the bright spots correspond to non-fully plasticized starch granules that kept their semicrystalline structure [2].

The PLM results confirmed our previous findings [2,15,31] that single-step SC does not yield highly homogeneous starch. The non-fully plasticized granules could be observed in TPS without AH treatment (Figure 1a), and even more in the TPS after 60 min of AH (Figure 1b). The most homogeneous starch was obtained with the two-step SC+MM preparation protocol without AH treatment (Figure 1c). The two-step preparation after 60 min of AH resulted in less homogeneous material (Figure 1d). This could be explained by combining two facts known from the previous studies: (i) the amorphous regions of starch granules are more susceptible to AH than the crystalline regions [23,43], and (ii) if the amorphous matrix is more degraded and less viscous, the disintegration of starch granules is less complete due to the lower shear forces during SC+MM processing [2]. Interestingly, an analogous trend is observed in immiscible polymer blends: if the viscosity of the matrix decreases, the disintegration of minor phase droplets during the melt mixing is less complete and the structure coarsens [44,45].

The morphological changes were monitored by SEM as well. The SEM/SE micrographs of TPS fracture surfaces (Figure A1 in Appendix B) confirmed the results of PLM (Figure 1), but the differences among the samples could not be observed so clearly. The not-fully plasticized starch granules exhibited higher contrast in polarized light (where they could be distinguished clearly as bright spots due to their anisotropic nature) than in SEM/SE micrographs (where they could be observed only in the form of blunt, rounded asperities on fracture surfaces).

#### 3.1.2. Vibrational Spectroscopy

Figure 2 summarizes the vibrational spectroscopy results for all 12 studied samples, i.e., the three starch types (original powder, TPS after SC, and TPS after SC+MM) with four AH times (0, 15, 30, and 60 min). Both infrared spectroscopy (Figure 2a) and Raman scattering (Figure 2b) were in agreement that the dominant chemical change was the incorporation of glycerol into the starch structure during SC, while AH and MM did not alter the starch molecular structure significantly (see also Figure A2 in Appendix B). The peak positions in TPS and their assignment to characteristic vibrations of starch and glycerol corresponded to those in the detailed study of Almeida et al. [46].

There was a clear difference between the spectra of all original starch powders (glycerol-free materials) and the spectra of all TPSs (with glycerol added during SC). The change in the spectra could be attributed to the fact that the glycerol molecules penetrated the granules and interacted with starch molecules, forming new intra- and intermolecular hydrogen bonds [47]. Within each group of materials (powders, TPS after SC, and TPS after SC+MM), negligible variations were observed with the AH time. We conclude that AH caused starch chain scissions, which changed both the morphology (as discussed above) and properties (as discussed below), but the concentration of the newly formed end groups was below the detection limit of IR and Raman spectroscopy. This is in accordance with the literature [48], even if Chung et al. [49] have demonstrated that near-infrared (NIR) spectra are more sensitive in this case, enabling the monitoring of the extent of AH. The differences between spectra of all TPSs after SC and after SC+MM were insignificant, although there were some local variations (we note that each spectrum in Figure 5 is a normalized average of >150 individual spectra, as described in the Experimental Section).

#### 3.1.3. Wide-Angle X-Ray Scattering

Figure 3 shows the results of wide-angle X-ray scattering (WAXS) of all investigated samples as a function of increasing acid hydrolysis time. The native starch powders (Figure 3, upper row) possessed the highest crystallinity. TPS after the one-step SC protocol (Figure 3, middle row) exhibited slightly lower crystallinity due to partial disintegration of native starch granules after gelatinization. TPS after the two-step SC+MM protocol (Figure 3, lower row) showed the lowest crystallinities due to (almost) complete destruction of the starch granules after the melt mixing. With an increasing acid hydrolysis time, the crystallinity slightly increased (see all three rows of Figure 3 from left to right). This is in accordance with the literature, which documents that the AH of starch preferentially targets the amorphous regions, enhancing crystallinity and double-helical content [23,25]. The crystallinity values for our investigated samples, ranging from 14 to 30%, are largely in line with other available reports [29,50]. The increase in crystallinity after chain scissions in amorphous region is a general trend that has been observed also for other semicrystalline polymers, such as ultra-high-molecular-weight polyethylene (UHMWPE) after oxidative degradation [51,52].

All starch powder diffractograms (Figure 3, upper row, all AH times) exhibited prominent diffraction peaks around 2θ ≈ 15°, 17°, 18°, and 23° at the CuKα wavelength. Moreover, several lower-intensity peaks could be observed at higher angles around 27°, 31°, 33°, 38°, and 41°. All above-listed peaks are characteristic of the starch type-A crystallinity, which is typical of cereals [25,29,30,53]. Therefore, the results documented that AH alone attacked mostly the amorphous regions and did not change the starch crystallinity type. Nevertheless, after SC, the crystallinity started to change. Some diffractions decreased and/or disappeared due to partial destruction of the original amylopectin A-type crystallinity. Some new diffraction appeared, corresponding to newly formed amylose V-type crystals [2,53] and amylopectin B-type crystals formed due to A-to-B-type crystallinity transition [53,54]. After SC+MM, the crystallinity changed substantially. The diffractions of the original starch powder either disappeared completely or merged into broad peaks in the regions of 15–25° and 30–45°. The only sharp diffractions at 13.5°, 20°, and 21° corresponded to V-type crystallinity [2,50]. A table summarizing all observed diffraction peaks together with a figure showing selected diffraction patterns with annotated diffraction are given in Appendix B (Table A1 and Figure A3).

### 3.2. Mechanical and Rheological Properties

The final TPS samples after SC+MM with various degrees of AH were characterized in detail by thermomechanical (Figure 4), rheological (Figure 5), and micromechanical measurements (Figure 6 and Figure 7). The three methods were in agreement that the properties of all samples were quite similar, although a slight increase in stiffness with increasing AH time was observed. The similar properties of the final TPS samples could be explained as follows: (i) the AH treatment caused chain scissions in the amorphous region, (ii) the chain scissions led to a softer and less viscous amorphous phase, (iii) the lower viscosity of the amorphous phase resulted in less complete disintegration of the crystalline phase during solution casting and melt mixing as documented by the PLM and WAXS results above, and (iv) the two contradictory effects—the softer amorphous phase with shorter molecules and the higher volume fraction of the less disintegrated crystalline phase—tended to cancel out, although the impact of increased crystallinity prevailed moderately at the end. The increase in TPS stiffness after AH was observed in previous studies as well [30,55].

#### 3.2.1. Dynamic Mechanical Thermal Analysis

Figure 4 displays the thermomechanical properties of the final TPS samples after SC+MM. The data were obtained from oscillatory shear rheometry (Section 2.4.4) as temperature sweeps (*T* from −90 to 140 °C) at a constant frequency (1 Hz). All four samples showed almost identical behavior. For the storage modulus (*G*′; Figure 4a) and loss modulus (*G*″; Figure 4b), the differences are hard to differentiate in the logarithmic scale plots. Nevertheless, the insets in Figure 4a,b document that acid hydrolysis tended to increase the values of *G*′ and *G*″ slightly in the region of a laboratory temperature around 20 °C. This agreed quite well with the results of micromechanical measurements (Figure 6 and Figure 7).

The damping factors (tan δ; Figure 4c) of all four samples exhibited three peaks around −50 °C, +30 °C, and +75 °C. These peaks corresponded to multiple glass transition temperatures (*T*_g_) of TPS, as explained in detail elsewhere [2]. Briefly, the lowest-temperature peak is linked to the highly plasticized regions with a high concentration of low-molecular-weight plasticizers (occasionally called *plasticizer-rich phase*), the intermediate peak is linked to less plasticized regions with a higher concentration of starch molecules (occasionally called *starch-rich phase*), and the highest-temperature peak can appear if the starch-rich phase contains fragments of stiff semicrystalline structures. The exact shape of the DMTA curves depends on the starch type, plasticization protocol, additives, humidity, aging, etc., but we can conclude that our results were in reasonable agreement with previous studies [15,56,57]. More details concerning the interpretation of DMTA curves of TPS can be found in the abovementioned work of Rana et al. [2] and references therein. Complete DMTA results are given in the Appendix A.

#### 3.2.2. Rheological Properties

Figure 5 shows the rheological properties of the final TPS samples after SC+MM. The data were obtained from oscillatory shear rheometry (Section 2.4.5) as frequency sweeps (angular frequency, ω, ranging from 0.1 to 100 rad/s) at a constant temperature (120 °C). The selected measurement temperature corresponded to the *nominal* TPS processing temperature, which was preset before the melt mixing (Section 2.3.2). The extremely high molar mass of starch molecules meant that all samples exhibited a behavior typical of crosslinked polymers in the whole frequency range [17]: (i) *G*′ > *G*″ in the whole range (solid and stiff network), and (ii) |*η**| decreasing monotonously from “infinitely high” values (no zero-shear viscosity plateau typical of linear polymers with common *M*).

In analogy with DMTA, the frequency sweeps in Figure 5 confirmed the similar behavior of all TPS samples, regardless of the AH time. The insets in each plot of Figure 5 show the minute differences between the samples at angular frequency ω = 2π rad/s, which corresponded to the frequency *f* = 1 Hz that was employed in DMTA (reminder: ω = 2π*f*). The comparison of the insets in the temperature sweep plots (Figure 4a,b) and frequency sweep plots (Figure 5a,b) confirmed that AH slightly increased the TPS resistance to elastic deformation (*G*’ increasing with AH time in the insets of Figure 4a and Figure 5a), while the TPS resistance to viscous flow was influenced much less (*G*″ exhibiting similar values and no clear trends in the insets of Figure 4b and Figure 5b). The absolute value of complex viscosity |*η**| showed a moderate increase with AH time (Figure 5c). As AH was expected to decrease the viscosity of TPS, this result is somewhat counter-intuitive. The simple explanation is that |*η**| is proportional to both *G*’ and *G”* (reminder: |*η**| = |*G**|/ω = (*G*′^2^ + *G*″^2^)^1/2^/ω) and *G*′ increased with AH reproducibly, while *G*″ values did not change significantly. The more detailed explanation follows in Section 3.3, which compares the TPS behavior in model conditions (an oscillatory flow in a rheometer) and in experimental conditions (a real flow during the melt mixing in a laboratory kneader).

#### 3.2.3. Micromechanical Properties

Figure 6 and Figure 7 summarize the results of instrumented microindentation hardness testing (MHI) measurements for the four final TPS samples after SC+MM preparation. The micromechanical properties assessed from MHI can be divided into two groups [39,40]: (i) stiffness-related properties, such as indentation modulus, *E*_IT_, indentation hardness, *H*_IT_, and Martens hardness, *H*_M_, and (ii) viscosity-related properties, such as indentation creep, *C*_IT_, and the elastic part of the indentation work, *η*_IT_. The main results connected with the stiffness-related properties are shown in Figure 6, the main results linked with viscosity-related micromechanical properties are shown in Figure 7, and the exact definitions of all above-listed micromechanical properties can be found in Appendix C (Figure A4).

The MHI measurements of stiffness-related properties (Figure 6) confirmed that AH enhanced the overall stiffness of TPS. The clear monotonous increase with AH time was observed for *E*_IT_ (Figure 6a) as well as for *H*_IT_ and *H*_M_ (Figure 6b). This was in accordance with both DMTA (Figure 4a) and rheological measurements (Figure 5a), in which another stiffness-related property, *G*’, increased with AH time as well. The trends in MHI (Figure 6) look clearer and more reproducible than in the case of DMTA and rheology (Figure 4 and Figure 5). This resulted from one important advantage of micromechanical measurements: Due to the small size of indentations (typically around 100 μm), we can perform tens or even hundreds of measurements with one specimen. In our case, the temperature sweeps (Figure 4) represent averages of 2 independent measurements, the frequency sweeps (Figure 5) are averages of 9 experiments, and the micromechanical properties (Figure 6 and Figure 7) originate from more than 90 independent measurements, as explained in Section 2.4.6. Another confirmation of the reliability and reproducibility of the MHI measurements is displayed in Figure 6c. Both theoretical considerations [58,59] and previous experimental results [39,60] suggest that many polymer systems exhibit approximate linear correlations between all stiffness-related properties. In our case, the theoretically predicted linear correlation between *E*_IT_ and *H*_IT_ was almost perfect, as documented in Figure 6c.

The viscosity-related properties (Figure 7) were affected by AH less than the stiffness-related properties. This corresponded to DMTA and rheological measurements (Figure 4 and Figure 5), in which the viscosity-related loss moduli, *G*″, were influenced less than the stiffness-related storage moduli, *G*′. The values of *C*_IT_ and *η*_IT_ only slightly decreased and increased with AH time, respectively. These shifts can be attributed to the fact that AH increases crystallinity, and as TPSs with higher contents of the crystalline phase are stiffer (higher *E*_IT_ and *H*_IT_), more resistant to creep (lower *C*_IT_), and more elastic (higher *η*_IT_). Analogous behavior was observed for other semicrystalline polymers, such as polypropylene [61] or poly(lactic acid) [37]. Figure 7c re-confirms the correctness of microindentation measurements, as the creep exhibits the typical power-law behavior, i.e., the typical linear increase in log(*h*^2^)-log(*t*) scale, as explained elsewhere [62]. Moreover, the values of creep constants, *n*, which were determined from fitting the power law to experimental data, are in good agreement with the literature [62,63].

### 3.3. Processing of Acid-Hydrolyzed Starches

Figure 8 shows the results of in situ measurements during the melt mixing of TPS. The melt mixing was the final preparation step of our thermoplastic starches, which resulted in the most homogeneous materials, as illustrated in Figure 1. The main objective of this work was to verify if the acid hydrolysis can decrease the processing temperature during the final melt mixing step. We note that the *nominal processing temperature*, which is preset in the laboratory kneader before the mixing (120 °C, as described in Section 2.3.2), somewhat increases after filling the mixing chamber with TPS due to the internal friction and energy dissipation [2]. As a result, during melt mixing, we can measure the energy needed to make TPS flow in the chamber (TQ, torque, moment of force; Figure 8a,b) and the increased *real processing temperature* (*T*; Figure 8c,d), as described in Section 2.4.7.

Figure 8 confirms that the AH could decrease both torque and processing temperature. The observed decrease was moderate for both quantities (Figure 8a,c), but even a few degrees can save a considerable amount of energy in large-scale applications [64,65] or improve the persistence of temperature-sensitive admixtures, such as antibiotics [31,33]. The TQ-*t* curves (Figure 8b) and *T*-*t* curves (Figure 8d) exhibited a steep change when TPS was added to the chamber during the first 2 min: TQ increased from zero (as we need enough energy to mix the cold and stiff material), and *T* decreased from the nominal 120 °C (as the source TPS, which was kept at ambient temperature, cooled the mixing chamber). After ca. 6 min of mixing, the TPS systems achieved a steady state (a plateau on TQ-*t* and *T*-*t* curves), where TQ decreased (the heated and homogenized material required less energy to flow) and *T* increased (a dissipation of energy due to internal friction of the flowing viscous material in the filled chamber). The average values of TQ and *T* in Figure 8a and Figure 8c were taken from the plateau regions of the TQ-*t* and *T*-*t* curves in Figure 8b and Figure 8d, respectively.

If we compare the rheological behavior of TPS in the model conditions during oscillatory shear rheometry (Figure 5) with the TPS behavior during the real melt mixing in the kneader (Figure 8), we can see a discrepancy that illustrates the limitations of simplified rheological measurements. In the rheometer, all TPS samples behaved as a *solid material* (Figure 5a,b: *G*′ > *G*″ in the whole frequency range) and the AH time *increased* their viscosity (Figure 5c). In the kneader, all TPS samples behaved as *viscous liquid* (as they could flow in the chamber) and the AH decreased their viscosity (as evidenced by the decrease in TQ and *T* for the AH-treated samples with respect to non-modified TPS). The explanation of this seeming contradiction is as follows: (i) The oscillatory shear rheometer characterizes TPS at 120 °C in the linear viscoelasticity range (Section 2.4.5). (ii) At the constant temperature of 120 °C and moderate oscillatory flow, the material keeps its structure and behaves as a solid. (iii) The kneader forces TPS into continuous movement, which increases the temperature due energy dissipation and disrupts many molecular entanglements together with remaining semicrystalline agglomerates. (iv) Under these conditions, the material starts to flow and behaves as a viscous liquid. (v) Once the material is under continuous flow, the shorter molecules present due to the AH start to act as lubricants that decrease the overall viscosity and the final processing temperature. We conclude that in our specific case of TPS with extremely long molecules and semicrystalline aggregates, the simple oscillatory shear rheometry could allow a basic characterization of rheological behavior (as summarized in Figure 5), but it could not yield a realistic prediction of the rheological behavior during the real melt mixing experiment (as documented in Figure 8).

## 4. Conclusions

The main objective of this work was to verify if the acid hydrolysis (AH) can decrease the processing temperature of thermoplastic starch (TPS) during the melt mixing. We subjected native wheat starch to 0, 15, 30, and 60 min of AH and converted the modified materials to highly homogeneous TPS by means of a two-step preparation protocol, which comprised solution casting (SC) followed by melt mixing (MM). All prepared TPSs were thoroughly characterized by multiple methods including polarized light microscopy (PLM), vibrational spectroscopy (infrared and Raman), wide-angle X-ray scattering (WAXS), dynamic mechanic thermal analysis (DMTA), oscillatory shear rheometry, instrumented microindentation hardness testing (MHI), and in situ measurements of torque and temperature during the final MM step. The key results are summarized below:The acid hydrolysis preferentially targeted the amorphous regions, which decreased the average molar mass and viscosity of the amorphous matrix. During the SC and MM, the lower-viscosity matrix in the AH-treated starches resulted in lower disruption of semicrystalline regions, a slightly coarser morphology (as observed by PLM), and higher crystallinity (as evaluated from WAXS).The fact that TPS after AH exhibited a less viscous, softer amorphous phase and higher fraction of stiffer crystalline phase resulted in similar thermomechanical, rheological, and micromechanical properties of all prepared systems (as evidenced by DMTA, rheometry, and MHI), because the two phenomena tended to cancel out.During the melt mixing, when the material was fully molten and forced to flow, the shorter molecules in all AH-treated TPSs started to act as a lubricant and decreased both torque (TQ) and processing temperature (*T*), as proven by in situ measurements in the kneading chamber during the MM. Even if the changes in TQ and *T* were quite moderate, we have demonstrated that AH is a feasible approach to save energy during TPS processing and/or to mitigate the negative effect of the MM on possible temperature-sensitive admixtures.

## Figures and Tables

**Figure 1 polymers-17-01310-f001:**
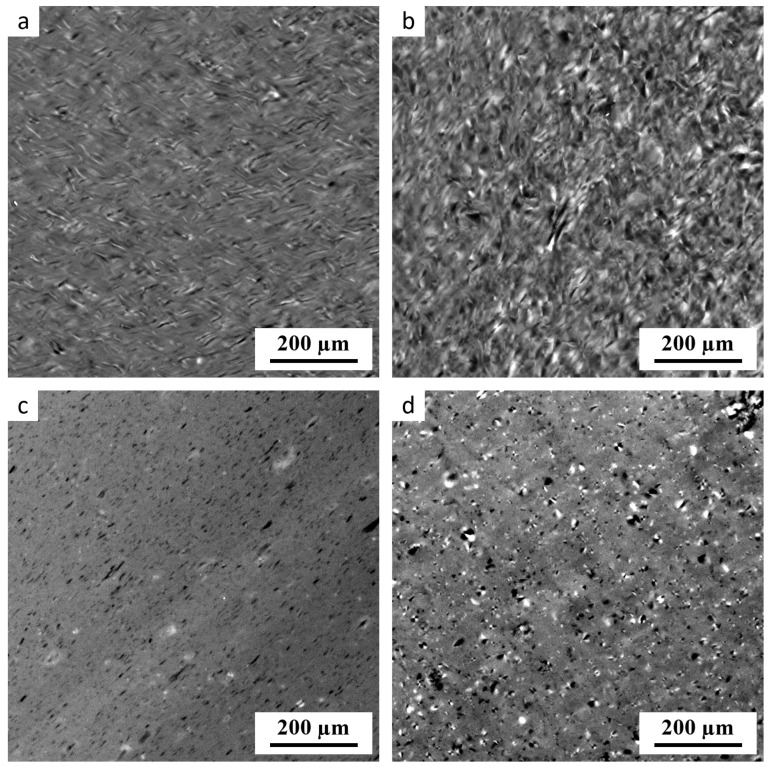
PLM micrographs showing four thermoplastic starch samples: (**a**) TPS-AH-0min after SC, (**b**) TPS-AH-60min after SC, (**c**) TPS-AH-0min after SC+MM, and (**d**) TPS-AH-60min after SC+MM. The samples without the AH treatment (**a**,**c**) contain fewer anisotropic inhomogeneities (bright spots) than the corresponding samples after AH (**b**,**d**). The complete list of the prepared TPS samples is given in Table 1.

**Figure 2 polymers-17-01310-f002:**
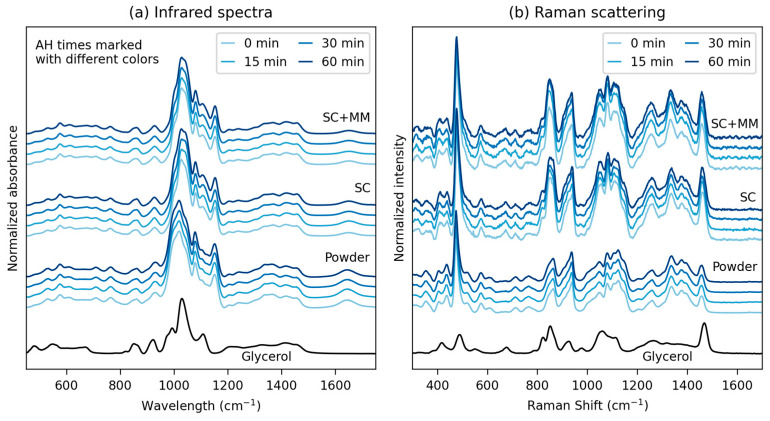
Vibrational spectroscopy results: (**a**) infrared spectra measured in ATR (attenuated total reflectance) mode and (**b**) Raman scattering measured at 785 nm excitation. Each of the two plots shows, from top to bottom, the following spectra: pure glycerol, original starch powders with various AH times, TPS after single-step SC preparation with various AH times, and TPS after two-step SC+MM preparation with various AH times.

**Figure 3 polymers-17-01310-f003:**
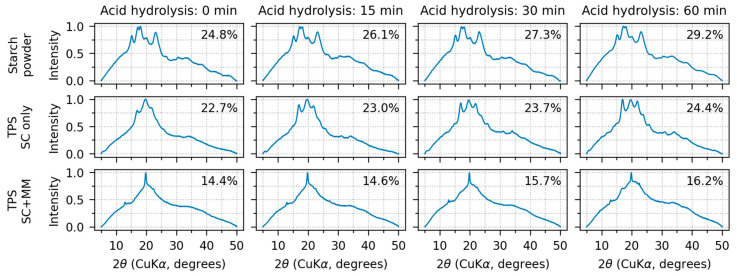
WAXS diffraction patterns and crystallinities of all investigated samples. The crystallinity values are printed in the upper right corner of each subplot. The rows show, from top to bottom, original starch powder, TPS after solution casting (SC), and TPS after solution casting and melt mixing (SC+MM). The columns show, from left to right, the studied materials after 0, 15, 30, and 60 min of acid hydrolysis.

**Figure 4 polymers-17-01310-f004:**
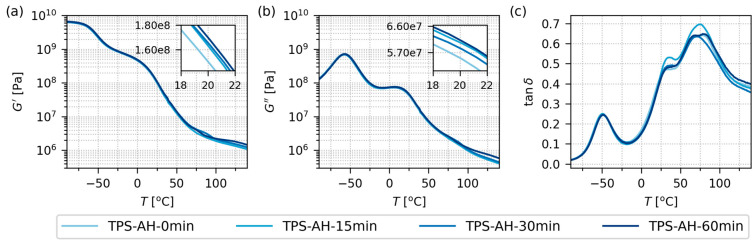
Thermomechanical properties of all investigated TPS samples after solution casting and melt mixing, measured by DMTA in rectangular torsion mode in the temperature range from −90 to 140 °C: (**a**) storage modulus, *G*′, (**b**) loss modulus, *G*″, and (**c**) damping factor, tan(*δ*). As the properties of all four samples were quite similar and hard to differentiate on a logarithmic scale, the insets in (**a**,**b**) show the properties at the laboratory temperature (20 °C) in greater detail. Note that *G*′ and *G*″ are plotted with the same y-axis limits in order to facilitate the direct comparison of the two quantities.

**Figure 5 polymers-17-01310-f005:**
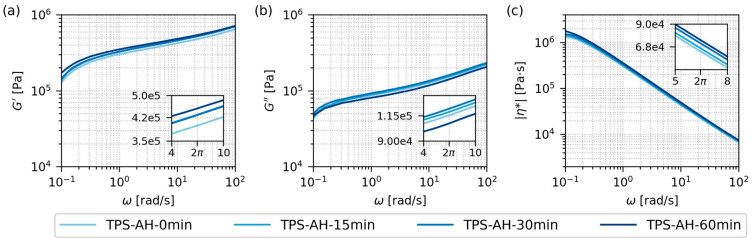
Rheological properties of all investigated TPS samples after solution casting and melt mixing, measured by oscillatory shear rheometry at 120 °C: (**a**) storage modulus, *G*′, (**b**) loss modulus, *G*″, and (**c**) absolute value of complex viscosity, |*η**|. As the properties of all four samples were quite similar and hard to differentiate on a logarithmic scale, the insets show the properties at angular frequency ω = 2π in greater detail. The angular frequency 2π corresponds to the frequency of 1 Hz that was employed in DMTA experiments (Figure 4; *f* = 2πω). *G*′ and *G*″ are plotted with the same y-axis limits in order to facilitate the direct comparison of the two quantities. Due to the extremely high molar mass of starch, the storage and loss modulus curves showed no intersection (*G*′ > *G*″ in the whole frequency range) and |*η**| exhibited a monotonous decrease (shear thinning without a plateau in a low shear range).

**Figure 6 polymers-17-01310-f006:**
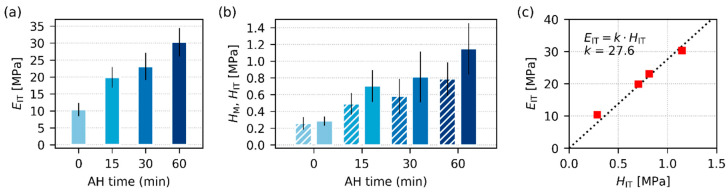
Stiffness-related properties of all investigated TPS samples after SC+MM, obtained from microindentation hardness testing: (**a**) indentation modulus, *E*_IT_, (**b**) Martens hardness (*H*_M_; striped columns) and indentation hardness (*H*_IT_, standard filled columns), and (**c**) the theoretically predicted linear correlation between *E*_IT_ and *H*_IT_. For each sample, we performed at least 90 indentations and averaged the results; the error bars represent estimated standard deviations.

**Figure 7 polymers-17-01310-f007:**
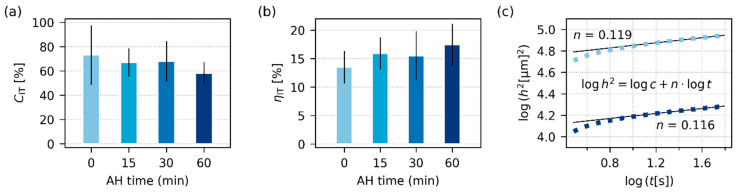
Viscosity-related properties of all investigated TPS samples after SC+MM, obtained from microindentation hardness testing: (**a**) indentation creep, *C*_IT_, (**b**) elastic part of the indentation work, *η*_IT_, and (**c**) the theoretically predicted linear correlation between the squared indenter penetration depth (*h*^2^) and time (*t*) on a log-log scale. In (**c**), the light blue squares represent the TPS-AH-0min sample, the dark blue squares represent the TPS-AH-60min sample, and the thin black lines are linear regression curves. For each sample, we performed at least 90 indentations and averaged the results; the error bars represent estimated standard deviations.

**Figure 8 polymers-17-01310-f008:**
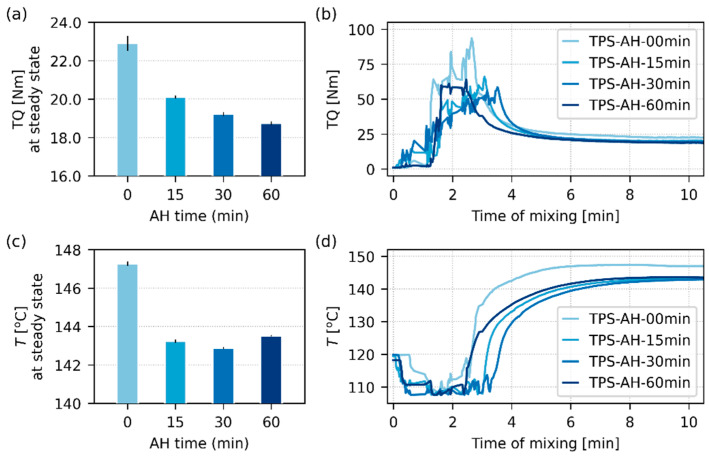
Processing parameters during the melt mixing of TPS-AH systems, which were measured in situ during the sample preparation: (**a**,**b**) torque (TQ) and (**c**,**d**) processing temperatures (*T*). The bar plots on the right (**a**,**c**) show the final average values of TQ and *T* in the steady state, i.e., after ca. 6 min of melt mixing. The line plots on the left (**b**,**d**) show TQ and T as a function of time during the whole melt mixing process. The mixing chamber was pre-heated to the *nominal* processing temperature of 120 °C, while the *real* processing temperature (*T*) was measured in situ as shown in (**d**); the real temperature increase was connected with the internal friction energy dissipation.

**Table 1 polymers-17-01310-t001:** List of prepared samples.

Native Starches *	Thermoplastic Starches **	AH Time (min) ***
S-AH-00min	TPS-AH-00min	0
S-AH-15min	TPS-AH-15min	15
S-AH-30min	TPS-AH-30min	30
S-AH-60min	TPS-AH-60min	60

* Native starches were all based on wheat starch type A, differing only in AH time. ** Thermoplastic starches were prepared by both single-step SC and two-step SC+MM protocols. *** Acid hydrolysis (AH) was applied to original native starches, from which TPSs were prepared.

## Data Availability

The data are available upon request to the corresponding author.

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
