# Peer review of "Impact of Acid Hydrolysis on Morphology, Rheology, Mechanical Properties, and Processing of Thermoplastic Starch"

_polymers, 2025, doi:10.3390/polym17101310_

Round 1
Reviewer 1 Report
Comments and Suggestions for Authors
Impact of acid hydrolysis on morphology, rheology, mechanical properties, and processing of thermoplastic starch
The paper presents interesting results regarding the influence of acid hydrolysis on physico-chemical properties of thermoplastic starch, such as morphology, crystallinity and mechanical properties. The paper provides well-documented experimental design and shows essential correlations concerning the influence of acid hydrolysis treatment on the properties of thermoplastic starch. On the other hand, some issues were found in data presentation and discussion of obtained results. After providing these minor revisions to the manuscript, the paper could be accepted for publication.
Specific remarks concerning the manuscript are given below:
- Line 39: give correct MW for amylose and amylopectin (106 and 108 g/mol).
- Sentence in lines 41-43 is unclear. What is do the authors mean under decomposition of starch and starch melting? Under what conditions (low water, excess water)? Please make clearer to the reader.
- Line 78, the phrase “yields the highly homogeneous starch”. Do the authors mean “yields the highly homogeneous thermoplastic starch”?
- Please give the value of amylose content for the starch sample used, since it is one of the main characteristics of starch, which influence its physico-chemical properties and susceptibility to acidic hydrolysis.
- The thickness of the obtained TPS samples was about 2 mm. The developed TPS would be subjected to further treatment like compression molding, etc., to obtain the final products like films, etc.? What possible applications could propose the authors for the obtained materials?
- The authors indicate that the storage conditions were 57% relative humidity. Is it possible to store the samples under ambient without the humidity control, which is not always suitable for real-life applications?
- Line 231. Some misprint was found (section Error! Reference 231 source not found.)
- Line 247 “confirmed that our previous findings” please correct to “confirmed our previous findings”.
- Could the authors give the estimations of the amount of the solubilized substance formed during acid hydrolysis of wheat starch? Kinetics of starch acid hydrolysis depends on the temperature, and it is important to compare the degree of AH in the present work with previously published results.
- Please increase the size of the PLM microphotographs in Figure 1 to show clearer the fine morphology details, and indicate the bright areas in the micrographs corresponding to the anisotropic material.
- Regarding FTIR and Raman spectroscopy results, it would be helpful to give the table, which summarizes the positions of the main FTIR and Raman lines, together with the attribution of the line positions to the vibrations of the particular chemical groups. This will give clear presentation of possible shifts in the positions of FTIR and Raman lines for the studied samples.
Also, from the Raman spectra of TPS it could be deduced that the peak positions for some lines (around 500 cm-1, in the 800-1000 cm-1 range) were shifted for SC+MM as compared to the SC samples. Please give the possible explanations.
- Please revise Figure 3 (WAXS patterns) in the same manner as Figure 2 (all curves on the single graph grouped and vertically shifted). This would help to present WAXS data more clearly. One could hardly see the details of the WAXS patterns on the small subplots. Also, please provide the method for the estimation of the crystallinity from the WAXS patterns.
- From the WAXS patterns of the TPS obtained using SC method it could be deduced that starch crystallinity type was changed from A-type for starch powder to B-type + V type after SC treatment. Moreover, prolonged AH tends to increase the B-type contribution to the WAXS pattern. This means that the amylopectin structure of the A-type in the starch powder during the SC step was melted (under the conditions of excess water wheat starch melts at 60-70 C range) and novel amylopectin structures of the B-type were formed during retrogradation of the samples. Acid hydrolysis promotes the formation of the B-type structures since there is increased amount of the short polysaccharide chains in the swollen gelatinized granules after AH and SC. Moreover, some amylopectin side chains could be also hydrolyzed from the AP backbone and form B-type structures in SC TPS samples. Therefore, the discussion of the effects of the AH on the SC type samples should be extended.
Also, please give some discussion why only V-type structures corresponding to amylose single helices were formed for SC+MM TPS types. Please also revise the Table A1 to give WAXS peaks corresponding to the A and B type starch structures.
Summing up, this is a good piece of work, providing interesting results on the preparation and physico-chemical properties of the thermoplastic starch. The paper could be accepted for publication after addressing the indicated issues in data presentation and discussion of the obtained results.

Author Response
Response to reviewer comments
We would like to thank both reviewers for their comments and questions, which helped us to improve the quality of our manuscript. Point-by-point answers to reviewer’s comments follow. The corresponding changes in the revised manuscript are marked with yellow background.
Reviewer #1
The paper presents interesting results regarding the influence of acid hydrolysis on physico-chemical properties of thermoplastic starch, such as morphology, crystallinity and mechanical properties. The paper provides well-documented experimental design and shows essential correlations concerning the influence of acid hydrolysis treatment on the properties of thermoplastic starch. On the other hand, some issues were found in data presentation and discussion of obtained results. After providing these minor revisions to the manuscript, the paper could be accepted for publication.
Specific remarks concerning the manuscript are given below:
Comment 1: Line 39: give correct MW for amylose and amylopectin (106 and 108 g/mol).
Response 1: Corrected. This was a formatting error. We thank the reviewer for careful checking.
Comment 2: Sentence in lines 41-43 is unclear. What is do the authors mean under decomposition of starch and starch melting? Under what conditions (low water, excess water)? Please make clearer to the reader.
Response 2: The sentence was rephrased. Now it reads: “Due to its high molecular weight, the native starch cannot be processed by melt-mixing like other common polymers, because it decomposes before melting.”
Comment 3: Line 78, the phrase “yields the highly homogeneous starch”. Do the authors mean “yields the highly homogeneous thermoplastic starch”?
Response 3: Yes, we mean “the highly homogeneous thermoplastic starch”. The sentence was corrected as the reviewer suggested.
Comment 4: Please give the value of amylose content for the starch sample used, since it is one of the main characteristics of starch, which influence its physico-chemical properties and susceptibility to acidic hydrolysis.
Response 4: The amylose content was around 25 %. This information was added to the revised manuscript (Experimental section, subsection 2.1).
Comment 5: The thickness of the obtained TPS samples was about 2 mm. The developed TPS would be subjected to further treatment like compression molding, etc., to obtain the final products like films, etc.? What possible applications could propose the authors for the obtained materials?
Response 5: The final samples were prepared by solution casting followed by melt mixing. In the very final step, the samples were compression molded to 2 mm thick plates. This thickness was selected because it was suitable for preparation of the testing specimens, namely for rheology (2 mm thick flat cylinders), DMTA (rectangular plates 40 mm x 10 mm x 2 mm), and microindentation (2 mm thick smooth surfaces prepared by microtomy). In an analogous way, the samples could be prepared in any other shape, suitable for a specific application. Some applications are in the last paragraph of the {Introduction} section. The end of the paragraph was slightly modified. It lists a few typical applications, such as biodegradable packaging, agriculture films, or pharmaceutics.
Comment 6: The authors indicate that the storage conditions were 57% relative humidity. Is it possible to store the samples under ambient without the humidity control, which is not always suitable for real-life applications?
Response 6: Yes, the TPS samples can be stored in open air. Nevertheless, storing the samples at well- defined humidity increases the reproducibility of the mechanical characterization methods. We added this information to the text of the revised manuscript (Experimental section, subsection 2.3.3, the end of the section).
Comment 7: Line 231. Some misprint was found (section Error! Reference 231 source not found.)
Response 7: The revised manuscript was corrected and contains no misprints around line 231.
Comment 8: Line 247 “confirmed that our previous findings” please correct to “confirmed our previous findings”.
Response 8: This was corrected as the reviewer suggested.
Comment 9: Could the authors give the estimations of the amount of the solubilized substance formed during acid hydrolysis of wheat starch? Kinetics of starch acid hydrolysis depends on the temperature, and it is important to compare the degree of AH in the present work with previously published results.
Response 9: As described in the {Experimental} section, we performed the acid hydrolysis according to ref. [32] for three different times (15, 30, and 60 min) and isolated the product. We did not try to estimate the amount of solubilized fraction as the reviewer suggested. Nevertheless, the extent of acid hydrolysis was increasing reproducibly with the hydrolysis time as evidenced by all methods (microscopy, rheology, mechanical properties, and processing properties).
Comment 10: Please increase the size of the PLM microphotographs in Figure 1 to show clearer the fine morphology details, and indicate the bright areas in the micrographs corresponding to the anisotropic material.
Response 10: The size of the PLM micrographs was increased. The bright areas were described better in the text (section 3.1.1; the end of the first paragraph) and in the legend of Figure 1. The changes in the are marked with yellow background.
Comment 11: Regarding FTIR and Raman spectroscopy results, it would be helpful to give the table, which summarizes the positions of the main FTIR and Raman lines, together with the attribution of the line positions to the vibrations of the particular chemical groups. This will give clear presentation of possible shifts in the positions of FTIR and Raman lines for the studied samples.
Response 11: The peak positions in vibration spectra and their assignment to characteristic vibration of starch and glycerol corresponded to those in the detailed study of Almeida et al. The reference to this study was added to the revised manuscript (section 3.2.1, end of the first paragraph).
Comment 12: Also, from the Raman spectra of TPS it could be deduced that the peak positions for some lines (around 500 cm-1, in the 800-1000 cm-1 range) were shifted for SC+MM as compared to the SC samples. Please give the possible explanations.
Response 12: No, the shifts between the vibrational spectra of SC and SC+MM samples were insignificant. We added a figure to Appendix A, which documents this clearly (the reference to the figure was added in the first paragraph of section 3.2.1).
Comment 13: Please revise Figure 3 (WAXS patterns) in the same manner as Figure 2 (all curves on the single graph grouped and vertically shifted). This would help to present WAXS data more clearly. One could hardly see the details of the WAXS patterns on the small subplots. Also, please provide the method for the estimation of the crystallinity from the WAXS patterns.
Response 13: The reviewer is right that the small individual plots in Figure 2 show rather the overall trend than the fine details of each spectrum. We decided to keep this concise figure in the main text, because it illustrates the main differences between the diffractograms concisely and clearly, but we added an additional figure (in the format suggested by the reviewer) to Appendix A (Fig. A3). As for the method of crystallinity determination, we employed the standard procedure in program Fityk. This information and relevant reference were added to the Experimental part (section 2.4.3).
Comment 14: From the WAXS patterns of the TPS obtained using SC method it could be deduced that starch crystallinity type was changed from A-type for starch powder to B-type + V type after SC treatment. Moreover, prolonged AH tends to increase the B-type contribution to the WAXS pattern. This means that the amylopectin structure of the A-type in the starch powder during the SC step was melted (under the conditions of excess water wheat starch melts at 60-70 C range) and novel amylopectin structures of the B-type were formed during retrogradation of the samples. Acid hydrolysis promotes the formation of the B-type structures since there is increased amount of the short polysaccharide chains in the swollen gelatinized granules after AH and SC. Moreover, some amylopectin side chains could be also hydrolyzed from the AP backbone and form B-type structures in SC TPS samples. Therefore, the discussion of the effects of the AH on the SC type samples should be extended. Also, please give some discussion why only V-type structures corresponding to amylose single helices were formed for SC+MM TPS types. Please also revise the Table A1 to give WAXS peaks corresponding to the A and B type starch structures.
Response 14: The reviewer is right here. In the original manuscript, we considered just the original A-type crystallinity of the powder starch and the newly-formed V-type crystallinity. The reason why the A-to-B-type crystallinity transition escaped our attention was that the peaks of A- and B-types crystallinity are overlapping. In any case, we modified both the main text (section 3.1.3) and added more information about the A-type, B-type, and VH-type crystallinity to the Appendix A (modified Table A1 and newly added Fig. A3).
Summing up, this is a good piece of work, providing interesting results on the preparation and physico-chemical properties of the thermoplastic starch. The paper could be accepted for publication after addressing the indicated issues in data presentation and discussion of the obtained results.
Reviewer 2 Report
Comments and Suggestions for Authors
Author investigated the impact of acid hydrolysis on morphology, rheology, mechanical properties and processing of thermoplastic starch, this provides important theoretical support for the industrial application of TPS, but there are still some parts of this manuscript that need further improvement.
1. Further standardize the writing of abstracts, such as briefly introducing the purpose and significance of the research in one sentence at the beginning of the abstract, instead of directly writing experimental methods.
2. Although the introduction provides a detailed review of the structure of starch, the preparation methods of TPS, and the application of acid hydrolysis, some of the content is somewhat lengthy and can be appropriately streamlined to improve the readability of the paper.
3. Acid hydrolysis is a common hydrolysis method, so in this study, the significance of using this method and its differences from other methods should be emphasized.
4. How is the time range of 0-60 minutes determined?
5. Add key parameter annotations (such as shear thinning zone and storage modulus intersection point) in the rheological performance diagram. Mark characteristic peaks (such as V-shaped crystal structures) with arrows or colors in XRD patterns to avoid relying solely on textual descriptions.
6. There are significant issues with the reference format of the article, such as the references in Line 231.
7. Are there specific process parameters for TPS hot pressing forming, such as holding time and cooling rate.
Author Response
Response to reviewer comments
We would like to thank both reviewers for their comments and questions, which helped us to improve the quality of our manuscript. Point-by-point answers to reviewer’s comments follow. The corresponding changes in the revised manuscript are marked with yellow background.
Reviewer #2
Author investigated the impact of acid hydrolysis on morphology, rheology, mechanical properties and processing of thermoplastic starch, this provides important theoretical support for the industrial application of TPS, but there are still some parts of this manuscript that need further improvement.
Comment 1: Further standardize the writing of abstracts, such as briefly introducing the purpose and significance of the research in one sentence at the beginning of the abstract, instead of directly writing experimental methods.
Response 1: The abstract was modified as the reviewer suggested. The third sentence gives the final objective of the study. The experimental methods are listed more briefly. The changed text is marked with yellow background.
Comment 2: Although the introduction provides a detailed review of the structure of starch, the preparation methods of TPS, and the application of acid hydrolysis, some of the content is somewhat lengthy and can be appropriately streamlined to improve the readability of the paper.
Response 2: Based on the comments of both reviewers, the {Introduction} was slightly modified. We managed to shorten the third paragraph (the longest one). We also modified the end of the last paragraph that describes the benefits of TPS with high homogeneity and low viscosity for selected applications. All changes are marked with yellow background in the revised manuscript.
Comment 3: Acid hydrolysis is a common hydrolysis method, so in this study, the significance of using this method and its differences from other methods should be emphasized.
Response 3: The answer to this question is given in the third and fourth paragraph of the {Introduction}. As already mentioned above, these two paragraphs were slightly modified based on the comments of both reviewers. The second reviewer is right that acid hydrolysis is a common method how to reduce starch molecular weight. Alternative methods how to decrease starch molecular weight are listed at the beginning of the 3rd paragraph, together with the relevant references. The novelty of our contribution is discussed at the end of the 3rd paragraph. The specific questions we asked in our contributions are summarized in the 4th (i.e. the last) paragraph of the {Introduction}. Briefly, we conclude that the acid hydrolysis is a well-established industrial process, particularly in the context of food applications, but limited information is available on its role in starch plasticization, namely its possible role in the decreasing of TPS processing temperature.
Comment 4: How is the time range of 0-60 minutes determined?
Response 4: The time range for acid hydrolysis (AH) was based on both available literature and our experiments. For example, Zhang et al. (ref. [30] in the revised manuscript) used AH times from 0 to 150 min, but our test suggested that AH times higher than 60 min did not yield homogeneous morphology and reproducible results. The investigation of longer AH times and other conditions during the acid hydrolysis (concentration and temperature) on the structure and processing properties of TPS are a subject of our ongoing research.
Comment 5: Add key parameter annotations (such as shear thinning zone and storage modulus intersection point) in the rheological performance diagram. Mark characteristic peaks (such as V-shaped crystal structures) with arrows or colors in XRD patterns to avoid relying solely on textual descriptions.
Response 5: We improved the description of rheology and WAXS experiments as the reviewer suggested. The rheology of TPS is rather specific due to its extremely high molecular weight. There is no storage and loss modulus intersection point (storage modulus is higher than loss modulus in the whole investigated range) and shear thinning zone spreads over the whole measured range (no plateau at low shear range displaying zero-shear viscosity and no plateau at high shear range displaying limiting high-shear viscosity). Therefore, we could not annotate those non-existing features in TPS curves, but we modified the text and figure legend in section 3.2.1. to make this clearer. As for WAXS, we improved the concise description in the main text (section 3.1.3) and we added more details to Appendix A. In the appendix, we improved the description of WAXS diffraction peaks in Table A1 (this was requested by the other reviewer as well) and we added new Figure A3 to the Appendix A (the figure shows selected WAXS diffraction pattern at higher detail with characteristic peaks marked).
Comment 6: There are significant issues with the reference format of the article, such as the references in Line 231.
Response 6: The reference issues around line 231 were corrected.
Comment 7: Are there specific process parameters for TPS hot pressing forming, such as holding time and cooling rate.
Response 7: The requested parameters are described in the Experimental section, subsection 2.3.2.
Round 2
Reviewer 2 Report
Comments and Suggestions for Authors
The article has been well revised. Regarding the issues in the initial draft, the author has responded well and made detailed revisions. It is recommended to accept it.